# Imaging quasiperiodic electronic states in a synthetic Penrose tiling

Laura C. Collins[1], Thomas G. Witte[1], Rochelle Silverman[1], David B. Green[1] & Kenjiro K. Gomes[1]

Quasicrystals possess long-range order but lack the translational symmetry of crystalline solids. In solid state physics, periodicity is one of the fundamental properties that prescribes the electronic band structure in crystals. In the absence of periodicity and the presence of quasicrystalline order, the ways that electronic states change remain a mystery. Scanning tunnelling microscopy and atomic manipulation can be used to assemble a two-dimensional quasicrystalline structure mapped upon the Penrose tiling. Here, carbon monoxide molecules are arranged on the surface of Cu(111) one at a time to form the potential landscape that mimics the ionic potential of atoms in natural materials by constraining the electrons in the two-dimensional surface state of Cu(111). The real-space images reveal the presence of the quasiperiodic order in the electronic wave functions and the Fourier analysis of our results links the energy of the resonant states to the local vertex structure of the quasicrystal.

[1] Department of Physics, University of Notre Dame, Notre Dame, Indiana 46556, USA. Correspondence and requests for materials should be addressed to K.K.G. (email: kgomes@nd.edu).

 1

The unique long range order of quasicrystals is evidenced by scattering[1]. Dan Shechtman first observed the 10-fold symmetric electron diffraction patterns from a rapidly cooled Al–Mn alloy[2]. Shechtman discovered a new type of material with long-range orientational and quasiperiodic order, but with rotational symmetries that cannot coexist with the translational symmetry found in classically defined crystals[3]. The sharp scattering peaks are a result of this quasiperiodic order, which can be visualized by the formation of moiré patterns. This order arises from the repetition of the same tiling patterns over and over again, with well-defined rules albeit in a non-periodic manner[3]. The electronic behaviour in periodic systems is dominated by extended wavefunctions while, in contrast, completely aperiodic or disordered systems are dominated by localized wavefunctions[4]. Quasicrystals tread right in the middle between these two conflicting limits, combining the lack of periodicity with the repetitions over long range, leading some studies to suggest critical wave functions which are neither localized nor extended[5].

The Penrose tiling is a classic example of quasiperiodic order, and it has been studied by mathematicians even before the discovery of quasicrystals[6]. Tight-binding studies for the Penrose vertex model quasicrystal, with the nearest neighbour edge hopping showed a largely degenerate state in the middle of the band containing almost 10% of all states in the band[7,8]. This degeneracy is typical for quasiperiodic tilings, and is due to families of strictly localized states[9] supported on a finite number of vertices, which reappear at various places throughout the tiling[10]. These localized states may be caused not by a disordered local potential of the sites, but instead by the local vertex structure that differentiates different sites[11]. The density of states of Penrose tiling quasicrystals is often described as spiky and presenting pseudogaps[12,13].

Photoemission measurements of natural quasicrystals have observed the existence of a pseudogap, but they are not able to resolve the fine structure of the density of states[14,15]. Tunnelling spectroscopic studies have also measured a pseudogap and they observe a spatially inhomogeneous local density of states, with signatures of electronic localization[16,17]. Even though high-quality imaging of quasicrystals has been achieved[18,19], many of the most pivotal theoretical predictions for the electronic properties in quasicrystals is yet to be to supported experimentally.

The use of synthetic quantum matter has been an extremely fruitful path to explore the physics behind natural systems. Synthetic quasicrystals have been modelled for cold atoms[20,21] and photonics[22]. In this experiment we realize a form of quantum simulation through the creation of artificial lattices[23,24]. To create a quasicrystal where we had unsurpassed control over the structure and imaging capabilities, we use the atomic manipulation capabilities of a scanning tunnelling microscope (STM)[25] to assemble the quasicrystal one molecule at a time and spectroscopic imaging to visualize the electronic density of states.

## Results

**Assembly and design of our synthetic quasicrystal.** Our experiment starts with a perfectly flat and clean Cu (111) surface, where we adsorbed carbon monoxide (CO) molecules. We used the STM tip to move each CO molecule into their correct positions prescribed by the rhombic Penrose tiling. This Penrose tiling is formed by the arrangement of two kinds of rhombi with the same side length but different vertex angles. One CO molecule is placed at the centre of each rhombus of the tiling, repelling the electrons to the vertices of the tiles and allowing flow along their edges. This kind of arrangement of electrons at the vertices of the tiles and hopping along their edges is known as the Penrose

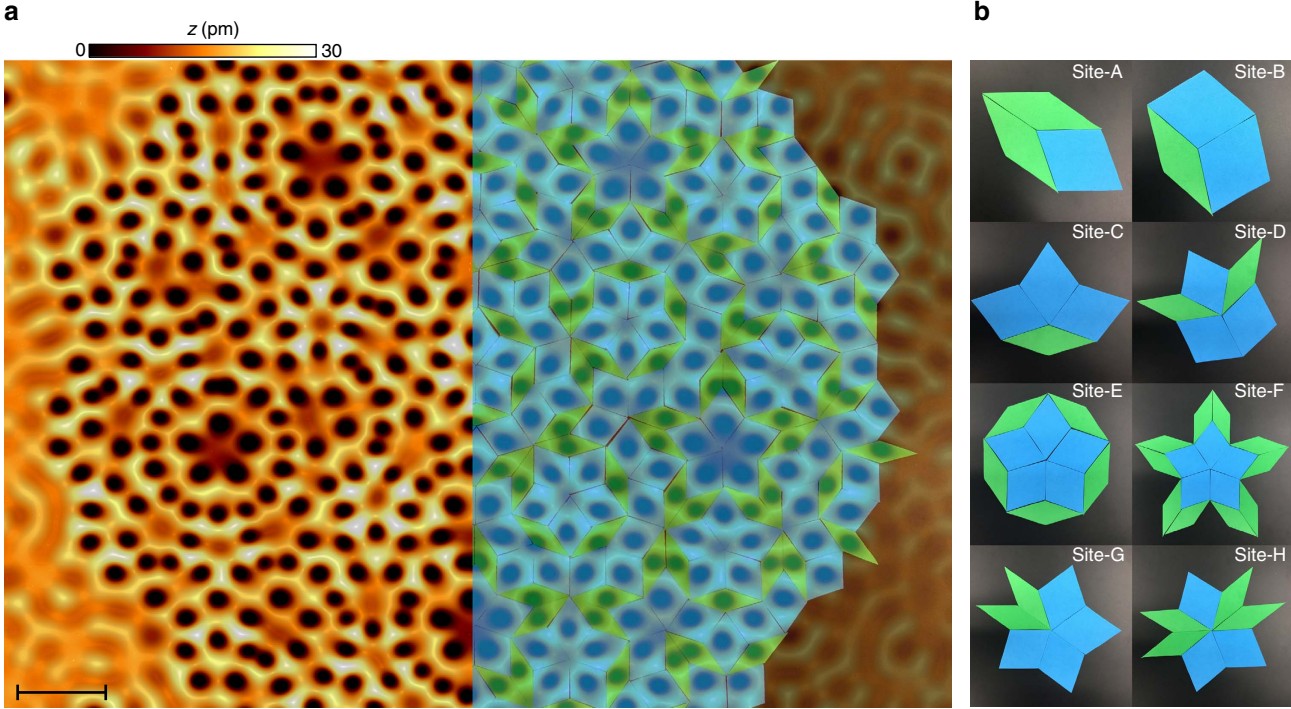

**Figure 1 | Synthetic Penrose tiling quasicrystal.** (**a**) STM topograph of assembled quasicrystal composed of 460 CO molecules measured at a bias voltage $V = 10$ mV and setpoint current $I = 1$ nA. The CO molecules are located at the centre of each dark spot in the topograph. The overlay on the right side is the Penrose tiling composed of rhombi with side length $a_0 = 1.6$ nm and vertices angles 72°/108° (blue) and 36°/144° (green). Scale bar, 4 nm. (**b**) Atlas of the eight types of vertex sites encountered in the Penrose vertex model tiling.

vertex model quasicrystal, and it has been extensively studied by tight-binding model calculations[26]. The STM topograph of the final assembled quasicrystal is shown in Fig. 1a, with the overlay showing the Penrose tiling. The completed quasicrystal is made up of 460 CO molecules, spaced such that the length of the side of each rhombus measures $a_0 = 1.6$ nm. Since the CO molecules register to the position of Cu atoms on the surfaces, their placement is not exact for this non-periodic arrangement, but the imprecision is always smaller than 0.15 nm.

The Penrose tiling forms eight structurally different sites[27], associated with each type of the first order vertex structure in the tiling, classified by the star combination of rhombi vertices. Interestingly, not every combination of rhombi is found in the Penrose tiling, as some configurations lead to spaces that cannot be extended. The eight types of sites allowed are shown in Fig. 1b, where the vertex at the centre of each image is the vertex site. In our experiment, we aim to weight the role played by the vertex types prescribed by the quasicrystalline structure in determining the local density of states. Because each vertex site is formed by a different arrangement of CO molecules, we may end up with slightly different local potentials for each vertex site. However, two of the sites shown in Fig. 1b, site-E and site-F, are formed by the same union of five 'fat' rhombi, but are classified as structurally different[6]. Even though they are locally identical at the first order, those sites are formed by different matching rules imposed by the global quasicrystalline order, and as a result they should exhibit different electronic behaviour.

The complete quasicrystal is displayed in Fig. 2a. The visualization of the electronic states is done through the measurement of the differential tunnelling conductance at different bias voltages, normalized by the spatially averaged tunnelling conductance measured on the bare Cu surface (see Methods for details). The normalized conductance map taken at the Fermi energy of the Cu surface states (bias voltage $V = 0$ mV) reveals an intriguing pattern (Fig. 2b), where electrons form a standing wave resonance but only at particular sites. The histogram of the normalized conductance sorted by the site type reveals that the electrons are predominantly localized on the B-sites (Fig. 2c). The diagram in Fig. 2d highlights the position of the B-sites and makes the observed pattern even clearer upon a direct comparison with the conductance map. At this particular energy, electrons resonate at a particular local vertex structure of the quasicrystal sites but the existence of this unique state right at the Fermi energy is not accidental.

**Electronic behaviour of our synthetic quasicrystal.** We have designed the tile size such that the position of the brightest peak in the Fourier transform of the Penrose tiling is equal to twice the Fermi wave-number ($k_F = 2.1$ nm$^{-1}$) of the electrons in the surface state of copper. In periodic lattices, this match corresponds to when the wave-vector first touches the edge of the Brillouin zone, transforming the band structure from electron-like to hole-like. This change in the curvature of the energy dispersion relation creates a nesting at the Fermi surface, which leads to a resonance in the density of states, which can be measured by tunnelling spectroscopy. Figure 2e is the Fourier transform of the conductance map shown in Fig. 2b. The outer ring originates from scattering of the electronic waves outside of the lattice and it has a radius of $2k_F$. Notice that it overlaps with the brightest peak of the Fourier transform of the model Penrose tiling (Fig. 2f). Quasicrystals do not have a well-defined Brillouin zone or dispersion relation but the Bragg peaks in the Fourier transform still correlate to the quasicrystalline order. The brightest peak is associated with the repetition of B-sites

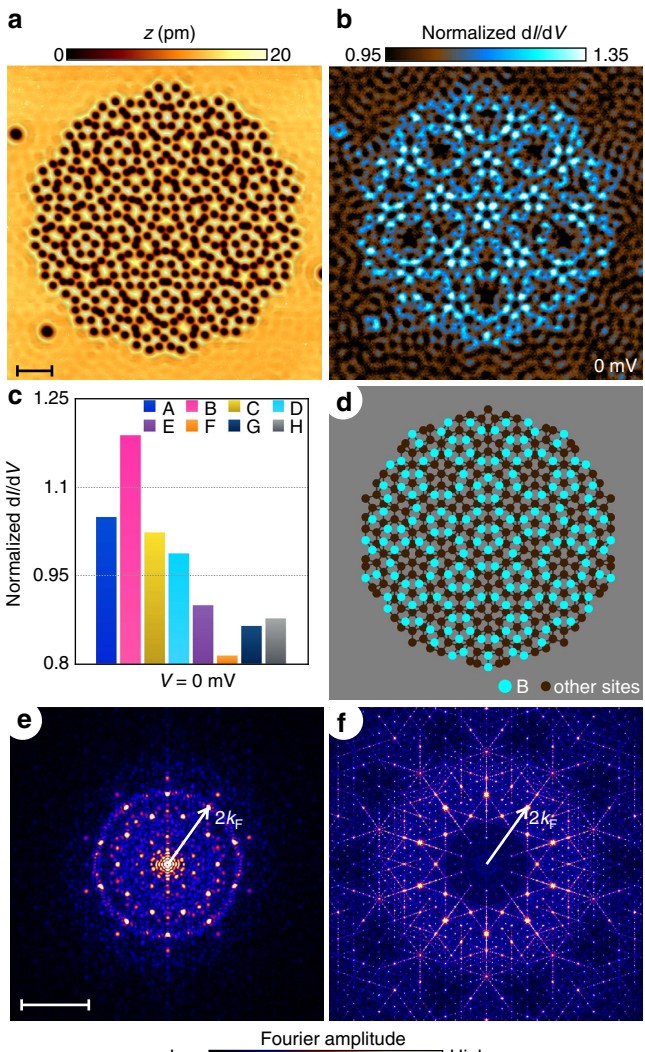

**Figure 2 | Visualizing a quasicrystalline electronic state. (a)** Topograph of the assembled quasicrystal composed of 460 CO molecules (dark spots in the image) measured with a bias voltage $V = 500$ mV and setpoint current $I = 500$ pA over a 42 nm by 42 nm field of view. Scale bar, 5 nm. **(b)** Normalized differential conductance map of the same field of view as **a** measured at bias voltage $V = 0$ mV. **(c)** Histogram of the normalized differential conductance at bias voltage $V = 0$ mV averaged over each type of vertex site. **(d)** Diagram of all vertex sites of the quasicrystal displaying the same field of view as **a** with site-B locations highlighted in cyan. **(e)** Fourier transform of the conductance map displayed in **b** with the arrow representing a wave-number value equal to twice the Fermi wave-number of the bare copper surface states ($k_F$). Scale bar, 4 nm$^{-1}$. **(f)** Fourier transform of quasicrystal model structure built with the same proportions as the one used in the experiment but with tens of thousands of sites to improve sharpness of Fourier peaks. The arrow represents the same wavenumber ($2k_F$) as in **e**.

(the most abundant of all site types) even though this repetition is non-periodic, and seems to result in our observed standing wave resonance at those sites. We show additional normalized conductance maps and their Fourier transforms in Supplementary Note 1 (Supplementary Fig. 1). To further clarify the connection between the quasiperiodic repetition of the B-sites and the brightest peak in the Fourier transform, we have taken the Fourier transform of the model Penrose tiling with only B-sites (Supplementary Fig. 2b) to examine the effects of their

quasiperiodic arrangement. We see the same brightest peak as the Fourier transform with the full model Penrose tiling (Fig. 2f), which indicates that the chosen tile size for our Penrose tiling indeed leads to a resonant state due to the quasiperiodic nature of the tiling. A more detailed explanation can be found in Supplementary Note 2.

While the structural differences between these sites and the connection to the quasicrystalline order is clear, tight-binding model calculations on the Penrose tiling have found that sites that are structurally similar can present different local density of states[9]. The electronic behaviour in our synthetic Penrose tiling is dictated by the Cu surface state electrons and their limited lifetime due to scattering to bulk states upon hitting the CO molecules. It is reasonable to assume that the electronic behaviour of a site is due to its first order vertex structure. To prove this nearest-neighbour limit in our system, we performed further analysis of the B-sites separated according to their second order vertex structure[27] (Supplementary Fig. 3) and there was a very small variance between the behaviour of the second order vertex structures. This analysis, along with a more detailed explanation can be found in Supplementary Note 2. Unlike tight-binding predictions[9] which predict the presence of forbidden B-sites at the Fermi energy, we see the same electronic behaviour from each B-site in our synthetic quasicrystal system. We must emphasize though that the electronic states are not being determined simply by the local potential created by the CO molecules. We can demonstrate this by showing how the E and F-sites have distinct electronic behaviour.

We measured conductance maps at different energies to further explore the dependence of resonant states in our synthetic Penrose tiling on the quasiperiodic arrangement of vertex sites. Conductance maps measured at bias voltage $V = -200$ mV (Fig. 3a) and $V = -100$ mV (Fig. 3b) show distinct patterns dominated by different site types. The map at $V = -200$ mV is dominated by the E, G and H-sites (Fig. 3c), while the map at $V = -100$ mV is dominated by the F-sites (Fig. 3d). The histograms of the conductance maps sorted by site type further demonstrate that pattern (Fig. 3e). On the basis of our comparison of the second order vertex structures of the B-sites (Supplementary Fig. 3) and the limited lifetime of the Cu surface state electrons, we assume the local potential of a site in our system is solely dependent on the arrangement of the nearest neighbours, so the E and F-sites have different local vertex structures but the same local potential since they are formed by the same pattern of five fat rhombi. We emphasize that even though the E and F-sites have the same local potential, the local density of states of these sites displays different energy dependence. The local density of states therefore depends on the arrangement of CO molecules beyond the first order vertex structure, which could reflect confinement effects from the second order vertex structure or the quasiperiodic arrangement of sites.

Our next step to corroborate the dependence of the density of states on the local vertex structure was to measure the differential conductance spectra over a large sampling of sites. We measured 1/5 of the vertex sites up to a radius of 12 nm (Fig. 4a). We avoided the edges of the sample to minimize the finite size effects. We normalized each differential conductance spectrum in the same manner as we did with the maps, by dividing each spectrum by the spatially averaged spectrum of the bare copper surface (Fig. 4b). We compare this normalization method to a few others in Supplementary Note 3 (Supplementary Fig. 4). The total normalized conductance spectrum (Fig. 4c) can be obtained by averaging all spectra and it displays a spiky morphology, as expected for quasicrystals[28]. The abundance of peaks in the Fourier transform results in an abundance of gaps and resonances in the density of states. We note that our synthetic system has

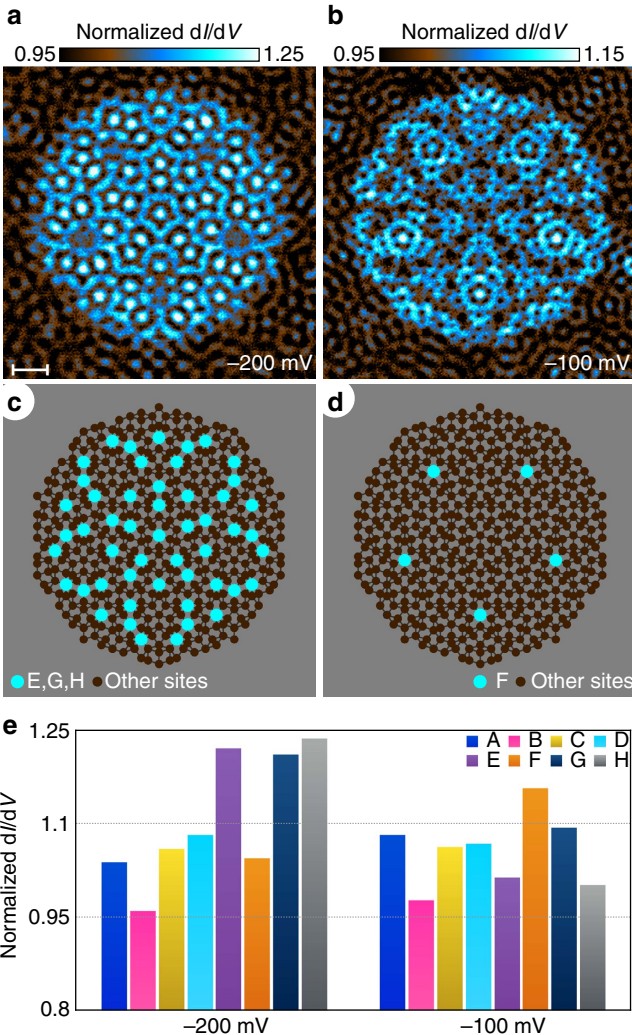

**Figure 3 | Topologically distinct sites.** (**a**) Normalized differential conductance map of a 42 nm by 42 nm field of view measured at bias voltage $V = -200$ mV. Scale bar, 5 nm. (**b**) Conductance map of the same field of view but at bias voltage $V = -100$ mV. (**c**) Diagram of all vertex sites of the quasicrystal displaying the same field of view as above and with site-E, site-G and site-H highlighted in cyan. (**d**) Diagram of all vertex sites of the quasicrystal displaying the same field of view as above and with site-F highlighted in cyan. (**e**) Histogram of the normalized differential conductance at bias voltage $V = -200$ mV (left) and $V = -100$ mV (right) averaged over each type of vertex site.

strong broadening effects in the conductance spectra because of the short lifetime of our surface state electrons due to scattering to the bulk states of Cu.

When comparing spectra taken at the same site types we noticed that they presented a very small variance between different locations. On the other hand, we noticed that spectra taken at different site types were consistently different, with a variance one hundred times larger than the variance of spectra taken at the same site types. In Fig. 4d, we present the spectra averaged by each site type. Notice that the sites with smaller areas (less distance between them and the nearest CO molecules in our system) show resonant states at higher energies than those with larger areas. These energy differences could be partially due to confinement effects in our synthetic Penrose tiling. However, since the E and F-sites have the same confinement at first order as

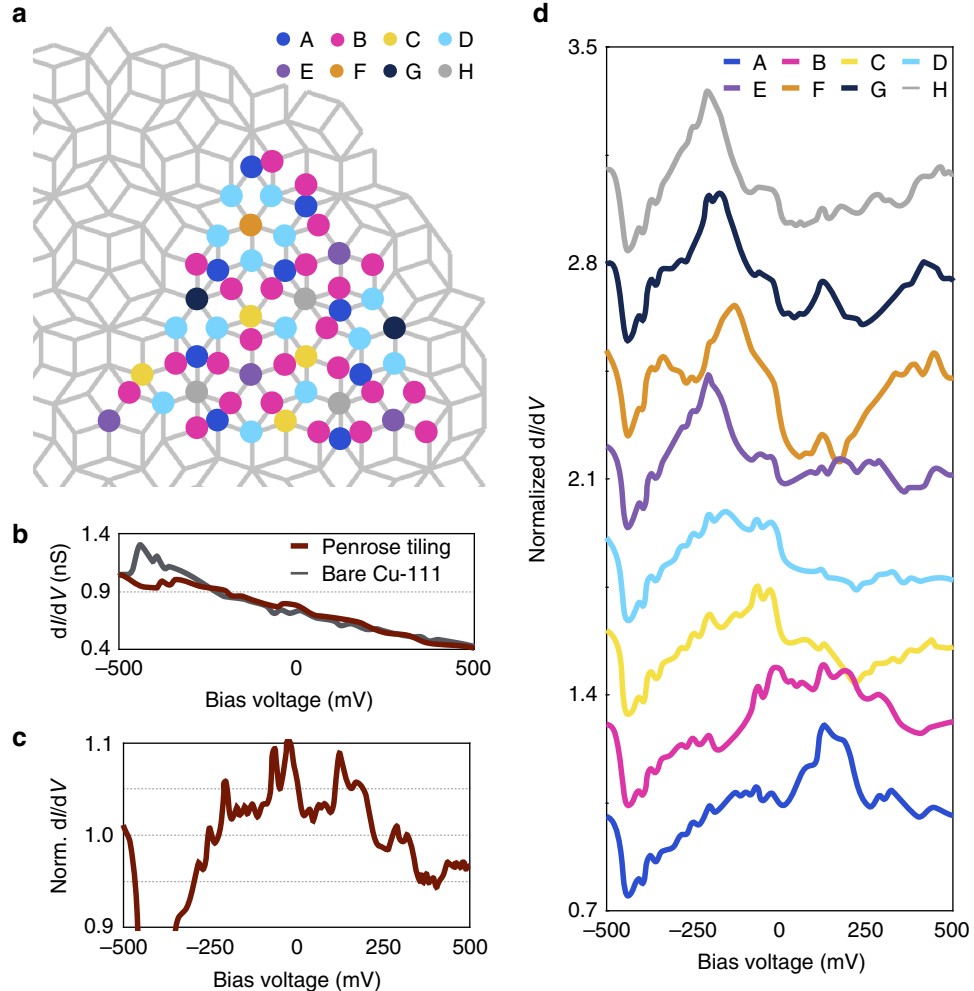

**Figure 4 | Tunnelling spectroscopy survey.** (**a**) The grey diagram illustrates the Penrose tiling assembled in our experiment. The colour dots mark the 61 electron sites where we measured the differential conductance spectra, with each colour corresponding to a site type. (**b**) The red line is the average of all 61 differential conductance spectra measure in the quasicrystal and the grey line a differential conductance spectra of the bare Cu (111) surface, spatially averaged over 100 distinct points. (**c**) The total normalized conductance spectra, calculated by the ratio of the two spectra presented above. (**d**) Normalized conductance spectra averaged by site type for all eight sites in the Penrose tiling. The y-scale refers to the bottom spectra taken at the A-sites. Other spectra have been offset for clarity by 0.3 units in the y-scale above the previous spectrum.

each other and exhibit different resonant states (Fig. 3), both the local vertex structure and quasiperiodic order of the system are important in determining the electronic states.

## Discussion

The creation of a synthetic Penrose tiling allowed for the first time to visualize the local density of state of a two-dimensional quasicrystal with sub-nanometer resolution. Our measurements have identified the existence of resonant states associated with Fourier peaks in reciprocal space and made a clear connection to the local vertex structure of each site. The extension of this experimental approach to include measurements of the changes in the electronic states due to scattering from defects could also solve other puzzles in the understanding of quasicrystals. Those measurements along with Fourier analysis should be able to reveal whether the electronic states are extended and dispersive or localized.

## Methods

**Sample preparation.** A single crystal Cu (111) sample was prepared by repeated cycles of sputtering (Ar ions, 1 keV) and annealing at a temperature of 1000 K in an ultra-high vacuum chamber. To deposit the CO molecules on the surface, we

briefly exposed it to $1 \times 10^{-8}$ torr partial pressure of CO gas, while keeping the sample temperature below 50 K.

**Normalized differential conductance spectroscopy.** All differential conductance spectra and maps were measure with a lock-in amplifier technique, with set point voltage $V = -500$ mV, set point current $I = 0.5$ nA and bias voltage modulation with root-mean-square average of $dV = 3.5$ mV. All energies are measured with the feedback loop open but the feedback loop is closed at the set point conditions to move between each point.

The differential conductance measured by a scanning tunnelling microscope is proportional to the density of states of the assembled structure, but is also proportional to the density of states of the STM tip, as well as the density of states of the bulk copper crystal. To isolate the density of states of the assembled lattice, we divided our measured conductance maps and tunnelling spectroscopy by the spatially averaged differential conductance of the bare surface of copper. The bare copper differential conductance was measured with the same tip conditions as the other measurements, and it is averaged over 100 distinct sites. This normalization minimized the contribution of varying tip conditions and the copper background and ensured that we mainly considered electronic contributions from our assembled quasicrystal. We also find that the normalized spectra are a much better fit to theoretical models of tunnelling spectroscopy, such as scattering theory.

**Data availability.** The data that support the findings of this study are available from the corresponding author on request.

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

## Author contributions

L.C.C. and K.K.G. conceived the project and designed the experiments. R.S. performed tight-binding calculations. T.G.W. and D.B.G. prepared the sample. T.G.W. and L.C.C. assembled the quasicrystal. L.C.C. and K.K.G. analysed the data and wrote the paper.

## Additional information

**Competing interests:** The authors declare no competing financial interests.

