## [Peer Review File · Nature Communications]

Reviewers' Comments:

Reviewer #1 (Remarks to the Author)

Review report: „Imaging Quasi-Periodic Electronic States in a Synthetic Penrose Tiling“ by L.C: Collins et al.

The authors report on the characterization of a Penrose Pattern assembled from individual CO molecules on Cu(111) by means of STM tip manipulation. The standing wave pattern created by the scattering of the 2D Cu(111) surface state electrons are measured by scanning tunneling spectroscopy and dI/dV mapping. The authors show the variation of the standing wave patterns with energy and attribute specific patterns to local structure configurations. By means of STS on equivalent first order vertex position of the Penrose pattern the authors show that the LDOS in this system depends on the type of local structure, but are similar for vertexes of the same first order structure type.

The method used by the authors has been used recently on different synthetic quantum mechanical bond structures to test theoretical predictions using an experimental “quantum simulator”. The authors place the CO molecule scatter in such way on the finite Penrose Pattern patch that in first approximation the conditions for an edge hopping Penrose vertex model is fulfilled. This theoretical model has been extensively studied in the past. The application of this experimental method on a quasi periodic structure is highly novel and interesting. The experimental data is of very good quality and well presented. However, the discussion of the observations remains on a phenomenological level and is in my view not adequate for a publication on the level of Nat. Comm., particularly in view of the extensive theoretical work done on this system. I believe that a more careful discussion of the experimental findings is needed to warrant a possible publication in Nat. Comm. In the following I will detail my comments and suggestions.

1. Abstract

“...periodicity is one of the fundamental properties that describe the electronic phenomena.” This is a very generic and not quite accurate statement. In molecules e.g. periodicity is not fundamental to understand the electronic structure. The authors most probably think of the importance of translational symmetry (periodicity) to describe electronic structure in terms of a band structure in crystals resp. in terms of Bloch waves. The authors should be more precise in describing what they mean here.

2. Abstract

The authors should be careful with the use of the term “topology” as it refers to a property which is invariant under a continuous deformation. The use here is rather to term the “local vertex structure”.

3. Page 3

“Tight-binding studies...showed a largely degenerate state.” This is not generally true. This state doesn't exist in all TB cases, but particularly for the nearest neighbor edge hopping vertex model (i.e. no hopping on the tile diagonal). This model is actually the one mimicked by the structure of the CO scatterers. The authors should clarify this point. Further down this paragraph again the term “topology” might not be well chosen.

4. Page 4

“eight topologically different electron sites” I think this statement is somewhat misleading. I would agree that there are 8 structurally different first order vertex points. However, according to TB simulations, first order vertex points of the same type (e.g. B) can be electronically very different. E.g. with respect to the degenerate $E=0$ state there allowed (non zero amplitude of the wave

function) and forbidden (zero amplitude for all $E=0$ state) B-sites. It is therefore not correct to classify them as "electronically equivalent" as I would understand the statement of the authors. The authors should clarify their understanding of "electronic sites".

5. Methods

The authors should give the details of the dI/dV measurement. Was a lock-in used, if yes what AC voltage amplitude and frequency. Where the dI/dV maps measured at constant height or in closed-loop?

6. Page 5

The authors have chosen the spacing of the CO scatterers such that the main resonance would occur at E_f ($U=0V$) for the Cu(111) SS. Here we would expect two properties in the total DOS i) a gap and the degenerate state in the mid gap. The authors should discuss if they observe these features. Actually the fact that the LDOS is similar on all B-sites seems to indicate otherwise, as we should have allowed and forbidden B-sites, which show a very different LDOS at $U=0V$ (resp. at the resonance).

7. Page 6

The authors refer to the application of a free electron like model where "we observed a resonant state similar to those found in crystals but displaying the quasi-periodicity...". The authors should give more details on these calculations (put them in the Sup Mat.) or make appropriate reference,

8. Page 6

The authors make reference to the high dimensional crystallography for quasi periodic structures. There are some inaccuracies. First, the Penrose Pattern can be generated from the projection of a slice of a 5-dim periodic lattice (resulting in a finite point density). It is the reciprocal lattice, which is a full (weighted) projection of the 5-dim periodic reciprocal lattice (resulting in an infinite point density).

The rationale of the statement "The quasi-periodic order emerges from this projection and is reflected in our findings" is not clear to me. Also the conclusion from the project to "We conclude that the existence of this resonant state... .. will be also strongly connected to the local topology of each site." Is not clear to me. The authors should carefully rewrite this paragraph or even consider to omit it.

9. Page 6

"...correlation between local topology and the quasi-periodicity". Due to the self-similarity and inflationary nature of quasi-periodic structures, there is no clear distinction between local structure and quasi-periodicity. E.g. the Fibonacci sequence LSLLSLSL can be understood from the local structures (LS) \rightarrow L' and (L) \rightarrow S' arranged in the sequence L'S'L'L'S' or from the local structures (LSL) \rightarrow L' and (LS) \rightarrow S' arranged on the sequence L'S'L'. The authors might refer more specifically to the order (e.g first order) of the vertex structures they consider.

Figure 3

It would be interesting to have the FT of the dI/dV maps, e.g. in the sup mat.

Figure 4 / Page 7

"spiky total DOS" looking at the total DOS of fig 4b it seems that many of the sharp peaks (around -20 meV and +100 meV) in fig 4c originate from features of the bare Cu(111) and are therefore a result of the normalization. The authors should carefully check this. Also it would be good to have the curves of 4b subtracted with a linear background and blown up (e.g. as graph in the sup. Mat.).

10. Page 7

The authors state that there is little variation of the LDOS for vertex types to the same first order. I fully agree with the authors that their finding support this statement. However, it is also the

point which raises my biggest concern and the requirement for a deeper discussion.

In the attached jpeg file there are some figures explaining my point. I have taken a similar Penrose patch as used by the authors and tried to simulate the dI/dV pattern by a simple superposition of circular, short ranged waves of different wave vectors paced at the tile centers (i.e. the position of the CO scatters in the experiment). The results approach the patterns seen experimentally to some extent, although there is no multiple scattering what so ever. I have also included the TB pattern of the degenerate $E=0$ state for the edge hopping vertex model. Of course this comparison is not very deep, however it raises the question what type of electronic coupling in the Penrose lattice the CO model of the authors represents. I have the suspicion that the CO model is significantly different from the vertex TB model. The reasons why this is the case are certainly complex and require extensive modeling, which I do not require from the authors. However, they might have given a hint with the limited lifetime of the SS electrons. This would translate in a considerable decoherence (loss of phase) in the tight-binding model and therefore make the LDOS dependent on local (close to first order) structures only. The downside of this interpretation is that the influence of the aperiodic structure might not be well tested in this CO model. However, there is also an upside in the sense that the influence of decoherence can be tested. Now this is speculative and I do expect the authors to go that far in the discussion in the paper. However, I think it is crucial to discuss that the degree of locality the authors observe is to some point at odds with the TB model they try to mimic. This might point to a better understanding of the CO scattering model, which I would find a highly interesting message and well worth of a Nat. Comm. publication.

Reviewer #2 (Remarks to the Author)

This is a very interesting manuscript that touches on fundamental issues of solid state physics, by exploring the connection between quasiperiodic atomic arrangement and electronic structure. By constructing a synthetic model of a quasicrystalline structure, the authors do groundbreaking work to understand this connection. Ever since the discovery of quasiperiodic alloys the question of "where are the electrons and how do they move" has been at the center of theoretical and experimental investigations of these unusual materials. The paper is thus of interest to a wide audience not even restricted to physicists, chemists, and materials scientists. As outlined below, I do have a number of points that should be clarified and several basic questions about the experiment and its interpretation. Moreover, there are some places in the manuscript where the text is misleading or inappropriate. Once these issues are dealt with, I believe the paper will be suitable for publication in Nature Communications.

- The action of the CO molecules in influencing the Cu(111) surface state electrons is not explained. When the authors say "...CO molecule is placed at the center of each rhombus repelling electrons to the vertices... and allow flow along the edges.." – how do they know? Is this commonplace knowledge?

- More on that point when looking at the Penrose tiling (Figure 1), it appears to me that the CO molecules are not located at the vertices of the tiling, but rather at the centers of the rhombi. What then constitutes the different "electron sites" – the absence rather than the presence of a CO molecule?

- The analysis of the data in Figure 2 (p. 5, 2nd paragraph) is cloudy. $2 \times k_F$ of the surface state is 0.4 rec. Angstroms (Kevan, Phys. Rev. Lett. 50, 526 (1983))

- please give a scale on the plot in Figure 2e so the reader can verify that the Brillouin zone boundary coincides with this value. The Cu(111) surface state (not the entire band structure as quoted in the text) thus changes from electron-like to hole-like at 0.2 rec. Angstroms. Hence there is no nesting, hence no resonance in the DOS.

- "Quasicrystals don't have a well defined Brillouin zone or dispersion relation" – but you show the "quasi-Brillouin zone" (Smith & Ashcroft Phys. Rev. Lett. 59, 1365 (1987), Niizeki & Akamatsu, J. Phys. Cond. Mat. 2, 2759 (1990) right there in Figure 2

- Generally, the observation of dispersive, almost free-electron-like bands (e.g. Rotenberg et al., Nature 406, 602 (2000), Rogalev, Nature Comm. 6, 8607 (2015) is entirely ignored. It would be a particularly important message of the present paper if the origin of the "spiky" DOS (Widmer et al, ref. 16) and a possible signature of the dispersion of the Cu(111) surface state could be seen in a combined picture.

- The conductance spectra (Fig. 4) are very interesting, since here (if you follow the argument of the authors) a direct connection between sites and DOS can be made, however, these data are only discussed in passing ("Once again, this upholds the relevance..." – yes, but how?).

Two other points:

- p.3, end of 2nd paragraph: "largely degenerate state in the middle of the band" – what band, why largely? This text comes much too early in the text to be meaningful. In fact, the entire paragraph is difficult to understand in its position in the flow of argument.

- The entire paragraph on p. 6 beginning with "One way to interpret..." is highly speculative and should be omitted.

Reviewer #3 (Remarks to the Author)

This study uses an artificially created Penrose tiling of finite size to address electronic states in quasi-periodic structures. Specifically, scanning tunnelling microscopy (STM) manipulation is applied to position 460 CO molecules in a quasicrystal-like topology. Scanning tunnelling spectroscopy (STS) then visualizes the response of the electronic structure of the Cu(111) surface state. The scattering of the surface state electrons at the CO molecules introduces a characteristic spatial modulation of the local density of states (LDOS) that depends on the energy / the applied bias voltage. Presenting dI/dV maps at specific bias voltages, the authors relate the LDOS maps to topologically distinct sites. In addition, it is shown that adding up the ST spectra recorded on these distinct sites leads to a "spiky" total LDOS, characteristic for quasicrystals.

During the last decade, the exploration of the local electronic structure of quasicrystals by means of STM and STS provided relevant insight into the intricate properties of such systems. Some properties of electronic states however still are elusive. Recently, also two-dimensional quasi-periodic tilings came into focus. Thus, this study makes a timely contribution to an interesting field of research. Specifically, the structure assembled molecule-by-molecule facilitates the interpretation of the data and provides a well-defined model system – apart from adding a beautiful example to the "library" of artificial structures created by STM. In addition, the manuscript reads nicely.

Nevertheless, I doubt that the manuscript in its present form is suitable for publication in a high impact journal as Nature Communications. In my opinion, the study does not provide really new insights into the physics of quasicrystals. For example, the variation of the LDOS in space and energy – including a "spiky" morphology – was described previously in great detail and even related to macroscopic properties as electrical resistivity (see for example refs. 17, 18). One aspect of the manuscript that might be considered new is the connection of resonant states with the local topology of specific sites. However, as pointed out below, this finding seems not really surprising and the data analysis is limited to a descriptive level. Thus, I cannot recommend

acceptance of the manuscript in its present form. The authors should highlight the novel aspects of their work more clearly and provide a more detailed analysis / interpretation of the spectroscopic data.

In the following, I list some of the issues that might be addressed by the authors:

(1) On page 6, one reads "The local density of states depends mostly on the local topology of the site and their quasi-periodic arrangement." This dependence is the key message of this manuscript and might need some clarification. As I understand, the term "local topology" is used to describe the sites (Fig. 1b). Thus, for example every site-B has the same local topology. However, due to the quasicrystalline nature of the assembly, there are different local environments for sites-B (Fig. 4a). On page 7, the authors note that the "spectra taken at the same site types .. presented a very small variance". Doesn't this imply that the LDOS is dictated mainly by the very local topology, the nearest CO centers – and thus not by their quasi-periodic arrangement (inducing a different local environment, see (3))?

(2) The spectroscopic data (E.g., Fig 4d) is presented in a rather descriptive way. Can the findings be rationalized, for example by calculating the LDOS? On page 6, one reads "If we apply an almost-free electron model approach, with the addition of the quasicrystalline order, by achieving the same "nesting" condition between the wave vector of the electrons and the Fourier peaks, we observe a resonant state similar to those found in crystals but displaying the quasi-periodicity associated with that peak." But this information is not really shown. A theoretical description / modeling might be beneficial.

(3) Could the authors address the potential influence of confinement effects? For example, site-A featuring the smallest area (between CO's) shows the surface state resonance at the highest energy and site-E featuring a large area shows the surface state resonance at the lowest energy (Fig. 4d). Similar effects were for example reported for disordered adatom lattices interacting with surface state electrons (Phys. Rev. Lett. 93, 146805, 2004). The resulting LDOS modulation thus would only indirectly be related to the quasicrystalline nature of the structure. In the same picture, the "spiky" LDOS (Fig. 4c) might also be achievable by averaging individual Sites-A to H (not forming a quasi-crystalline arrangement).

(4) A comparison of the FFT of the dI/dV map at 0 mV (Fig. 2e) with the FFT of the Penrose structure (Fig. 2f) given by the CO centers shows good agreement. Thus, the spatial modulation in the dI/dV maps reflects mainly the positions of the scattering centers, while the sites influence the intensity. Are there distinct differences to the FFT's of Fig. 3a and Fig. 3b?

(5) The wording should be adapted in some cases. For example, the quasicrystal was not assembled "one atom at a time" as stated on page 4. Also the conclusion that the study "allowed for the first time to visualize the local density of state.. with atomic resolution" seems rather bold. The length scale of the LDOS variations clearly exceeds atomic dimensions. Also the last sentence "The expanding this experimental approach.." should be corrected and it would be nice to specify how this experimental approach could be expanded.

Referee #1

1. Abstract

“...periodicity is one of the fundamental properties that describe the electronic phenomena.” This is a very generic and not quite accurate statement. In molecules e.g. periodicity is not fundamental to understand the electronic structure. The authors most probably think of the importance of translational symmetry (periodicity) to describe electronic structure in terms of a band structure in crystals resp. in terms of Bloch waves. The authors should be more precise in describing what they mean here.

We altered the sentence to be more specific to Bloch waves: “In solid state physics, periodicity is one of the fundamental properties that prescribes the electronic band structure in crystals.”

2. Abstract

The authors should be careful with the use of the term “topology” as it refers to a property which is invariant under a continuous deformation. The use here is rather to term the “local vertex structure”.

We have taken the referee’s advice and used the term “local vertex structure” throughout the manuscript.

3. Page 3

“Tight-binding studies...showed a largely degenerate state.” This is not generally true. This state doesn’t exist in all TB cases, but particularly for the nearest neighbor edge hopping vertex model (i.e. no hopping on the tile diagonal). This model is actually the one mimicked by the structure of the CO scatterers. The authors should clarify this point. Further down this paragraph again the term “topology” might not be well chosen.

We clarified the cited studies in the main text as “Tight-binding studies for Penrose vertex model quasicrystals with nearest neighbor edge hopping”. The word topology was substituted by vertex structure.

4. Page 4

“eight topologically different electron sites” I think this statement is somewhat misleading. I would agree that there are 8 structurally different first order vertex points. However, according to TB simulations, first order vertex points of the same type (e.g. B) can be electronically very different. E.g. with respect to the degenerate $E=0$ state there allowed (non zero amplitude of the wave function) and forbidden (zero amplitude for all $E=0$ state) B-sites. It is therefore not correct

to classify them as “electronically equivalent” as I would understand the statement of the authors. The authors should clarify their understanding of “electronic sites”.

We changed “topologically” to “structurally” to be consistent with rest of the text. We are just using the term electronic sites as the location where the electrons will be found in the same way that we use the term atomic sites for natural crystals. We made a few changes to this paragraph to make it more clear.

5. Methods

The authors should give the details of the dI/dV measurement. Was a lock-in used, if yes what AC voltage amplitude and frequency. Where the dI/dV maps measured at constant height or in closed-loop?

We added a few sentences to the methods section to clarify this point.

“All differential conductance spectra and maps were measure with a lock-in amplifier technique, with set point voltage $V = -500$ mV, set point current $I = 0.5$ nA and bias voltage modulation with root-mean-square average of $dV = 3.5$ mV. All energies are measured with the feedback loop open but the feedback loop is closed at the set point conditions to move between each point.”

6. Page 5

The authors have chosen the spacing of the CO scatterers such that the main resonance would occur at E_f ($U=0V$) for the Cu(111) SS. Here we would expect two properties in the total DOS i) a gap and the degenerate state in the mid gap. The authors should discuss if they observe these features. Actually the fact that the LDOS is similar on all B-sites seems to indicate otherwise, as we should have allowed and forbidden B-sites, which show a very different LDOS at $U=0V$ (resp. at the resonance).

We would like to clarify how the spacing for the CO molecules was chosen. Quoting from the manuscript: “We have designed the tile size such that the position of the brightest peak in the Fourier transform of the Penrose tiling is equal to twice the Fermi wave vector ($kF = 2.1$ nm⁻¹) of the electrons in the surface state of copper”. We are relating the spacing to a feature in k -space. Besides, it is important to note that we don’t expect to have the same resonance as the one found in tight binding. First, in our quasicrystal the on-site potential for each site type is not identical due to the proximity to the CO molecules. Second, there is slight disorder coming from the approximate position of each CO. Most importantly, we have a strong broadening effect coming from the scattering to the bulk so we wouldn’t be able to measure small gaps.

7. Page 6

The authors refer to the application of a free electron like model where “we observed a resonant state similar to those found in crystals but displaying the quasi-periodicity...”. The authors should give more details on these calculations (put them in the Sup Mat.) or make appropriate reference,

We have added the appropriate references to the paragraph.

8. Page 6

The authors make reference to the high dimensional crystallography for quasi periodic structures. There are some inaccuracies. First, the Penrose Pattern can be generated from the projection of a slice of a 5-dim periodic lattice (resulting in a finite point density). It is the reciprocal lattice, which is a full (weighted) projection of the 5-dim periodic reciprocal lattice (resulting in an infinite point density).

The rationale of the statement “The quasi-periodic order emerges from this projection and is reflected in our findings” is not clear to me. Also the conclusion from the project to “We conclude that the existence of this resonant state... .. will be also strongly connected to the local topology of each site.” Is not clear to me. The authors should carefully rewrite this paragraph or even consider to omit it.

We have taken the advice of both reviewer 1 and 2 and we have decided to omit this paragraph from the manuscript.

9. Page 6

“...correlation between local topology and the quasi-periodicity”. Due to the self-similarity and inflationary nature of quasi-periodic structures, there is no clear distinction between local structure and quasi-periodicity. E.g. the Fibonacci sequence LSLLSLSL can be understood from the local structures (LS) \rightarrow L' and (L) \rightarrow S' arranged in the sequence L'S'L'L'S' or from the local structures (LSL) \rightarrow L' and (LS) \rightarrow S' arranged on the sequence L'S'L'. The authors might refer more specifically to the order (e.g first order) of the vertex structures they consider.

We rephrased the sentence in question to clarify the main point we are trying to make. it reads “We measured conductance maps at different energies to demonstrate that the electronic states are not being determined by the local site potential but instead by the global quasi-periodic order of the structure.” The important point is to put emphasis on the global order and we demonstrate that point using the Fourier analysis.

Figure 3

It would be interesting to have the FT of the dI/dV maps, e.g. in the sup mat.

We have created a supplement with the requested FT and a few other datasets that were not included in the paper.

Figure 4 / Page 7

“spiky total DOS” looking at the total DOS of fig 4b it seems that many of the sharp peaks (around -20 meV and +100 meV) in fig 4c originate from features of the bare Cu(111) and are therefore a result of the normalization. The authors should carefully check this. Also it would be good to have the curves of 4b subtracted with a linear background and blown up (e.g. as graph in the sup. Mat.).

We also have added this analysis on the supplemental material.

10. Page 7

The authors state that there is little variation of the LDOS for vertex types to the same first order. I fully agree with the authors that their finding support this statement. However, it is also the point which raises my biggest concern and the requirement for a deeper discussion. In the attached jpeg file there are some figures explaining my point. I have taken a similar Penrose patch as used by the authors and tried to simulate the dI/dV pattern by a simple superposition of circular, short ranged waves of different wave vectors placed at the tile centers (i.e. the position of the CO scatters in the experiment). The results approach the patterns seen experimentally to some extent, although there is no multiple scattering what so ever. I have also included the TB pattern of the degenerate $E=0$ state for the edge hopping vertex model. Of course this comparison is not very deep, however it raises the question what type of electronic coupling in the Penrose lattice the CO model of the authors represents. I have the suspicion that the CO model is significantly different from the vertex TB model. The reasons why this is the case are certainly complex and require extensive modeling, which I do not require from the authors. However, they might have given a hint with the limited lifetime of the SS electrons. This would translate in a considerable decoherence (loss of phase) in the tight-binding model and therefore make the LDOS dependent on local (close to first order) structures only. The downside of this interpretation is that the influence of the aperiodic structure might not be well tested in this CO model. However, there is also an upside in the sense that the influence of decoherence can be tested. Now this is speculative and I do expect the authors to go that far in the discussion in the paper. However, I think it is crucial to discuss that the degree of locality the authors observe is to some point at odds with the TB model they try to mimic. This might point to a better understanding of the CO scattering model, which I would find a highly interesting message and well worth of a Nat. Comm. publication.

We reference the tight-binding model because it is most studied model for system we studied. Our results differ from this model in various ways and we agree with the referee that it may not be the best way to reproduce our results and a scattering model calculation may do a better job. We do put quite a bit of emphasis on the importance of the local structure of each site at determining the density of states of the electrons. A detailed study of the scattering from the CO molecules is indeed of high interest and we hope we can contribute to it in the near future.

Referee #2

- The action of the CO molecules in influencing the Cu(111) surface state electrons is not explained. When the authors say "...CO molecule is placed at the center of each rhombus repelling electrons to the vertices... and allow flow along the edges.." – how do they know? Is this commonplace knowledge?

The use of CO molecules to confine and/or scatter electrons has been done multiple times in the past. The referee can refer to reference [2] to see the same technique being used in a similar way.

- More on that point when looking at the Penrose tiling (Figure 1), it appears to me that the CO molecules are not located at the vertices of the tiling, but rather at the centers of the rhombi. What then constitutes the different "electron sites" – the absence rather than the presence of a CO molecule?

The CO molecules don't provide the electrons that we are measuring. Those come from the Cu-111 surface states. The CO molecules repel the electrons which then concentrate at the vertices of the rhombi. Moreover, the CO blocks any hopping diagonally on the rhombi and only allows hopping along the edges.

- The analysis of the data in Figure 2 (p. 5, 2nd paragraph) is cloudy. $2 \times k_F$ of the surface state is 0.4 rec. Angstroms (Kevan, Phys. Rev. Lett. 50, 526 (1983))

That is correct. We added the value of k_F number to make it more clear. We obtain k_F from the wavelength of the surface states of copper at the bias voltage $V=0\text{mV}$.

- please give a scale on the plot in Figure 2e so the reader can verify that the Brillouin zone boundary coincides with this value. The Cu(111) surface state (not the entire band structure as quoted in the text) thus changes from electron-like to hole-like at 0.2 rec. Angstroms. Hence there is no nesting, hence no resonance in the DOS.

We added the scale bar to figure 2e as requested.

- "Quasicrystals don't have a well defined Brillouin zone or dispersion relation" – but you show the "quasi-Brillouin zone" (Smith & Ashcroft Phys. Rev. Lett. 59, 1365 (1987), Niizeki & Akamatsu, J. Phys. Cond. Mat. 2, 2759 (1990) right there in Figure 2

Thanks for those references we have added those to our manuscript. It is still true that you don't have a proper Brillouin zone and our results support the idea illustrated by those authors that you still have special points in k-space that affect the electronic states. We will avoid calling it a quasi-Brillouin zone to avoid confusion.

- Generally, the observation of dispersive, almost free-electron-like bands (e.g. Rotenberg et al., Nature 406, 602 (2000), Rogalev, Nature Comm. 6, 8607 (2015) is entirely ignored. It would be a particularly important message of the present paper if the origin of the "spiky" DOS (Widmer et al, ref. 16) and a possible signature of the dispersion of the Cu(111) surface state could be seen in a combined picture.

We added the reference suggested to our text. We do mention “almost free-electron” approaches to understand our results but we would like to say that we have not observed dispersive bands in our experiments. Maybe more experiments with quasiparticle scattering measurements could clarify this point but presently we don't have data to prove this point.

- The conductance spectra (Fig. 4) are very interesting, since here (if you follow the argument of the authors) a direct connection between sites and DOS can be made, however, these data are only discussed in passing (“Once again, this upholds the relevance...” – yes, but how?).

This concluding sentence is just a summary from the fact that even though different site types present different density of states, we find that the same site type will present the same density of states, independent of where it is located in the crystal.

- p.3, end of 2nd paragraph: “largely degenerate state in the middle of the band” – what band, why largely? This text comes much too early in the text to be meaningful. In fact, the entire paragraph is difficult to understand in its position in the flow of argument.

The paragraph gives brief account of the main results found by numerical models of the Penrose tiling. We have change the language to be a bit more specific in the paper. Now it reads: “Tight-binding studies for Penrose vertex model quasicrystals with nearest neighbor edge hopping showed a largely degenerate state in the middle of the band containing almost 10% of all states in the band.”

- The entire paragraph on p. 6 beginning with “One way to interpret...” is highly speculative and should be omitted.

We have taken the advice of both reviewer 1 and 2 and we have decided to omit this paragraph from the manuscript.

Reviewer #3:

(1) On page 6, one reads “The local density of states depends mostly on the local topology of the site and their quasi-periodic arrangement.” This dependence is the key message of this manuscript and might need some clarification. As I understand, the term “local topology” is used to describe the sites (Fig. 1b). Thus, for example every site-B has the same local topology. However, due to the quasicrystalline nature of the assembly, there are different local environments for sites-B (Fig. 4a). On page 7, the authors note that the “spectra taken at the same site types .. presented a very small variance”. Doesn’t this imply that the LDOS is dictated mainly by the very local topology, the nearest CO centers – and thus not by their quasi-periodic arrangement (inducing a different local environment, see (3))?

It is possible that the DOS is being determined by the local structure of the sites but the match between the resonant states and the peaks in the Fourier transforms also highlight the importance of the quasiperiodicity. In quasicrystals both local structure and the long range order are deeply connected and they cannot be separated.

(2) The spectroscopic data (E.g., Fig 4d) is presented in a rather descriptive way. Can the findings be rationalized, for example by calculating the LDOS? On page 6, one reads “If we apply an almost-free electron model approach, with the addition of the quasicrystalline order, by achieving the same “nesting” condition between the wave vector of the electrons and the Fourier peaks, we observe a resonant state similar to those found in crystals but displaying the quasi-periodicity associated with that peak.” But this information is not really shown. A theoretical description / modeling might be beneficial.

We added additional references on that paragraph to clarify the point.

(3) Could the authors address the potential influence of confinement effects? For example, site-A featuring the smallest area (between CO’s) shows the surface state resonance at the highest energy and site-E featuring a large area shows the surface state resonance at the lowest energy (Fig. 4d). Similar effects were for example reported for disordered adatom lattices interacting with surface state electrons (Phys. Rev. Lett. 93, 146805, 2004). The resulting LDOS modulation thus would only indirectly be related to the quasicrystalline nature of the structure. In the same picture, the “spiky” LDOS (Fig. 4c) might also be achievable by averaging individual Sites-A to H (not forming a quasi-crystalline arrangement).

We would like to direct the referee’s attention to Figure 3 and the analysis of that data in the text. Notice that even though Sites E and F have the same first neighbor CO pattern, they have different resonances. We agree that there is some effect coming from the local potential in each site but we also found that the quasiperiodicity plays an important role.

(4) A comparison of the FFT of the dI/dV map at 0 mV (Fig. 2e) with the FFT of the Penrose structure (Fig. 2f) given by the CO centers shows good agreement. Thus, the spatial modulation in the dI/dV maps reflects mainly the positions of the scattering centers, while the sites influence the intensity. Are there distinct differences to the FFT’s of Fig. 3a and Fig. 3b?

The peaks in the FFT are coming from position of the CO molecules but the circle observed (absent in the model calculation) is coming from the wavelength of the surface states of copper.

The main differences between the FFT of figure 3A and 3B is that since they were measured at different energies, the radius of the circle of the copper surface states will be different and match different Fourier peaks of the quasiperiodic structure.

(5) The wording should be adapted in some cases. For example, the quasicrystal was not assembled “one atom at a time” as stated on page 4. Also the conclusion that the study “allowed for the first time to visualize the local density of state.. with atomic resolution” seems rather bold. The length scale of the LDOS variations clearly exceeds atomic dimensions. Also the last sentence “The expanding this experimental approach..” should be corrected and it would be nice to specify how this experimental approach could be expanded.

We adjusted to “one molecule at a time” and “sub-nanometer”. We rephrased the final sentence as “The extension this experimental approach to include quasiparticle scattering measurements”

Reviewers' Comments:

Reviewer #1 (Remarks to the Author)

Referee report "Imaging Quasi-Periodic Electronic States in a Synthetic Penrose Tiling " by L.C. Collins et al.,

The authors have made adequate changes to most of the point I have raised in my first review. In view that the discussion is still rather generic and that the methods the authors present has been well established on other systems, I feel that the paper is borderline to qualify for publication in Nat. Comm..

There are however two points which need to be properly addressed before publication.

1. Page 4 / "The Penrose tiling form eight structurally different electronic sites..." which the authors support with Ref 27.

I do not agree with the view of the authors that the notion of electronic site can be used the same as structural site. The eight sites the authors denote belong to a structural classification of sites sharing a common feature in this class (the same first order vertex structure). If one speaks of electronic sites, one insinuates a classification according to a common electronic feature within the class, as I outlined this is not the case for these eight vertex sites. B vertex sites can be of different electronic nature. The authors must change this incorrect impression and best omit the term electronic site if this doesn't refer to a clear classification criterion

2. Page 6 / "If we apply an almost-free electron model approach...".

This statement is still ambiguous. It is not clear if the authors have carried out a nearly-free electron model calculation of their structure and observed nesting peaks, or if they generally discuss the results of such calculations in literature.

If they have made calculations, they should put them in the Sup. Mat.

If they have not made such calculations, then they should clarify like "Calculations using a nearly-free electron model have shown that..." followed by the corresponding references.

Reviewer #2 (Remarks to the Author)

I've read the revised manuscript and the replies of the authors to the comments of all reviewers. In my view they provide an adequate response so the paper appears ready for publication. I do have a one comment about the revised version of the manuscript, however. While the authors say on p 6 that " (we) demonstrate that the electronic states are no being determined by the local site potential but instead by the global quasiperiodic order", they contradict this statement by saying at the end of the same paragraph that "the local DOS depends mostly on the local vertex structure of the site", and on p.7 that "once again, this upholds the relevance of the local vertex structure in determining the electronic states". This contradiction should be clarified. In my view, what this actually says is that, on the basis of the data presented in the paper, a distinction cannot be made. I believe this is a very nice experiment that opens the door to examining quasicrystallinity in a model system, but overstating the results does nothing to enhance its merit.

Beyond this comment, there are a few expressions throughout the manuscript that, now that the manuscript is close to acceptance, could be thrown out: „visualize the DOS with unmatched clarity" – clarity maybe, but the origin remains cloudy, since, as the authors say at the end, they still cannot distinguish between localized and dispersive states, the latter in fact having been

demonstrated in the photoemission experiments which the authors try to put down earlier. A similar expression "unprecedented detail" in the abstract is equally fancy and an overstatement in my view.

There is one typo in the figure caption to Figure 1: right side instead of "ride", and at the end of the caption for Figure 2 it should read "tens of thousands of sites".

Reviewer #3 (Remarks to the Author)

I acknowledge that the authors considered some of the reviewers advice, improving the manuscript. And I'm again impressed by the scientific approach and the beauty of this artificial structure. Unfortunately however, the modifications included in the resubmissions seem somewhat superficial.

A relevant issue raised in my previous report was the locality of the electronic states (see comments (1) and (3)) that was also identified by reviewer 1 in two comments (9. & 10.) as crucial. In my opinion, the reply of the authors does not clarify this issue and no relevant changes were included in the manuscript to deepen the discussion. Despite the clarification of some terms, the wording partially still seems confusing (e.g., page 6: "We.. demonstrate that the electronic states are not being determined by the local site potential.." versus "The local density of states depends mostly on the local vertex structure of the site.."). The degree of locality is important for the rationalization of the spiky LDOS.

The answer of the author's to issue 2 "We added additional references on that paragraph to clarify the point" does not clarify or even address the point. Also from a stylistic point of view the resubmission seems somewhat overhasty. A grammatical problem mentioned in my previous report (5) is not corrected and the references are not formatted properly (e.g., ref. 29). The supplementary information seems not mentioned in the manuscript.

I agree with the author's conclusion that they visualized for the first time the local density of state of a two-dimensional quasicrystal with sub-nanometer resolution. But still doubt that this alone justifies publication in a high impact journal as Nat. Comm.(even if the data and the structure are beautiful). My requests to better highlight the new insights into the physics of quasicrystals provided by this study and to go beyond a purely descriptive level were not really considered.

Thus, my feeling is that the authors unfortunately missed a chance to considerably improve this interesting manuscript in the first revision to make it suitable for publication in Nat. Comm.. If the open issues are addressed – including for example a tentative discussion of the locality – one could consider the manuscript for publication in Nat. Comm., otherwise I suggest publication in a more specialized journal.

Reviewers' comments:

Reviewer #1 (Remarks to the Author):

Referee report “Imaging Quasi-Periodic Electronic States in a Synthetic Penrose Tiling ”
by L.C. Collins et al.,

The authors have made adequate changes to most of the point I have raised in my first review. In view that the discussion is still rather generic and that the methods the authors present has been well established on other systems, I feel that the paper is borderline to qualify for publication in Nat. Comm..

There are however two points which need to be properly addressed before publication.

1. Page 4 / “The Penrose tiling form eight structurally different electronic sites...” which the authors support with Ref 27.

I do not agree with the view of the authors that the notion of electronic site can be used the same as structural site. The eight sites the authors denote belong to a structural classification of sites sharing a common feature in this class (the same first order vertex structure). If one speaks of electronic sites, one insinuates a classification according to a common electronic feature within the class, as I outlined this is not the case for these eight vertex sites. B vertex sites can be of different electronic nature. The authors must change this incorrect impression and best omit the term electronic site if this doesn't refer to a clear classification criterion

While B vertex sites have been shown to be different electronically in tight-binding models, that is not the case in our synthetic quasicrystal. We have added several sentences and done further analysis which is included in the Supplementary Information to show that the B vertex sites behave the same electronically in our system. To illustrate this, we separated the sites based on their second order vertex structure, and found that the variance between the various second order vertex structures was very small (several orders of magnitude smaller) compared to the variance between the various first order vertex structures. The spectra at the B-sites, sorted by second order vertex structure, as well as illustrations depicting the different second order vertex structures are shown in Supplementary Fig. 3. While we have added this discussion to clarify the differences between our system and tight-binding models, we have also omitted the term electronic sites from the manuscript to avoid confusion.

2. Page 6 / “If we apply an almost-free electron model approach...”.

This statement is still ambiguous. It is not clear if the authors have carried out a nearly-free electron model calculation of their structure and observed nesting peaks, or if they generally discuss the results of such calculations in literature.

If they have made calculations, they should put them in the Sup. Mat.

If they have not made such calculations, then they should clarify like “Calculations using a nearly-free electron model have shown that...” followed by the corresponding references.

Since we did not perform almost free-electron model calculations ourselves we have made the decision to omit that statement from the manuscript. Instead, we try to illustrate the same connection between the quasi-periodicity and the brightest peaks in the model Fourier transform by including another Fourier transform of the model Penrose tiling with only B-sites present in the Supplementary Information. We reference this in the manuscript, and notice that the brightest peaks in this additional Fourier transform match the brightest peaks in the Fourier transform of our synthetic Penrose tiling as well as that of the model of the full Penrose tiling. Since the peaks line up exactly in all three cases, we conclude that the resonant state evidenced by these bright peaks is due to the quasi-periodic arrangement of the B-sites. We also include a Fourier transform of just a model Penrose tiling with just the A-sites to show that these peaks don't simply come from limiting the sites present in the Penrose tiling.

Reviewer #2 (Remarks to the Author):

I've read the revised manuscript and the replies of the authors to the comments of all reviewers. In my view they provide an adequate response so the paper appears ready for publication. I do have a one comment about the revised version of the manuscript, however.

While the authors say on p 6 that “ (we) demonstrate that the electronic states are no being determined by the local site potential but instead by the global quasiperiodic order”, they contradict this statement by saying at the end of the same paragraph that “the local DOS depends mostly on the local vertex structure of the site”, and on p.7 that “once again, this upholds the relevance of the local vertex structure in determining the

electronic states”. This contradiction should be clarified. In my view, what this actually says is that, on the basis of the data presented in the paper, a distinction cannot be made. I believe this is a very nice experiment that opens the door to examining quasicrystallinity in a model system, but overstating the results does nothing to enhance its merit.

To clarify this, we changed the first sentence of the paragraph to

“We measured conductance maps at different energies to further explore the dependence of resonant states in our synthetic Penrose tiling on the quasi-periodic arrangement of electronic sites.”

where the changed part is italicized. We also attempted to clarify the conclusion of the paragraph that the local vertex structure plays more of a role in the local density of states than the local potential of a site (Site E vs. Site F). We conclude that the LDOS depends on the local vertex structure and the quasi-periodic arrangement of sites, which should clear up the apparent contradiction. While a distinction cannot be made between the local vertex structure and the quasi-periodic order, we believe we have made a definitive distinction between the local vertex structure and the local site potential.

Beyond this comment, there are a few expressions throughout the manuscript that, now that the manuscript is close to acceptance, could be thrown out: „visualize the DOS with unmatched clarity“ – clarity maybe, but the origin remains cloudy, since, as the authors say at the end, they still cannot distinguish between localized and dispersive states, the latter in fact having been demonstrated in the photoemission experiments which the authors try to put down earlier. A similar expression “unprecedented detail” in the abstract is equally fancy and an overstatement in my view.

We have removed these phrases.

There is one typo in the figure caption to Figure 1: right side instead of “ride”, and at the end of the caption for Figure 2 it should read “tens of thousands of sites”.

We have corrected these typos.

Reviewer #3 (Remarks to the Author):

I acknowledge that the authors considered some of the reviewers advice, improving the manuscript. And I'm again impressed by the scientific approach and the beauty of this artificial structure. Unfortunately however, the modifications included in the resubmissions seem somewhat superficial.

A relevant issue raised in my previous report was the locality of the electronic states (see comments (1) and (3)) that was also identified by reviewer 1 in two comments (9. & 10.) as crucial. In my opinion, the reply of the authors does not clarify this issue and no relevant changes were included in the manuscript to deepen the discussion. Despite the clarification of some terms, the wording partially still seems confusing (e.g., page 6: "We.. demonstrate that the electronic states are not being determined by the local site potential.." versus "The local density of states depends mostly on the local vertex structure of the site.."). The degree of locality is important for the rationalization of the spiky LDOS.

To clarify this, we changed the first sentence of the paragraph to

"We measured conductance maps at different energies to further explore the dependence of resonant states in our synthetic Penrose tiling on the quasi-periodic arrangement of electronic sites."

where the changed part is italicized. We also attempted to clarify the conclusion of the paragraph that the local vertex structure plays more of a role in the local density of states than the local potential of a site (Site E vs. Site F). We conclude that the LDOS depends on the local vertex structure and the quasi-periodic arrangement of sites, which should clear up the apparent contradiction. While a distinction cannot be made between the local vertex structure and the quasi-periodic order, we believe we have made a definitive distinction between the local vertex structure and the local site potential.

To further deepen the locality discussion, we have added several references to the limited lifetime of the surface state electrons in our system to illustrate its limitations. We also refer to first order vertex structures specifically when we discuss local electronic behavior with the exception of a comparison of the electronic behavior of the second order vertex structures for the B-sites in the Supplementary Information (also referenced in the main text). We compare the behavior of the B-sites with greater detail in the

Supplementary Information in an effort to further illustrate the differences between our system (where we do not see signatures of “forbidden B-sites”) and tight-binding models (where there is evidence of “forbidden B-sites”). We also added a brief reference to the effects of the sizes of the sites (possible confinement effects) on the electronic behavior.

The answer of the author’s to issue 2 “We added additional references on that paragraph to clarify the point” does not clarify or even address the point.

Since we did not perform almost free-electron model calculations we have made the decision to omit that statement from the manuscript. Instead, we try to illustrate the same connection between the quasi-periodicity and the brightest peaks in the model Fourier transform by including another Fourier transform of the model Penrose tiling with only B-sites present in the Supplementary Information. We reference this in the manuscript, and notice that the brightest peaks in this additional Fourier transform match the brightest peaks in the Fourier transform of our synthetic Penrose tiling as well as that of the model of the full Penrose tiling. Since the peaks line up exactly in all three cases, we conclude that the resonant state evidenced by these bright peaks is due to the quasi-periodic arrangement of the B-sites. We also include a Fourier transform of just a model Penrose tiling with just the A-sites to show that these peaks don’t simply come from limiting the sites present in the Penrose tiling.

Also from a stylistic point of view the resubmission seems somewhat overhasty. A grammatical problem mentioned in my previous report (5) is not corrected and the references are not formatted properly (e.g., ref. 29). The supplementary information seems not mentioned in the manuscript.

We have proofread more thoroughly this time, and corrected the grammatical problem that was not fixed before. We fixed the formatting of the references. We have also added more analysis figures to the Supplementary Information as well as several references to it throughout the manuscript.

I agree with the author’s conclusion that they visualized for the first time the local density of state of a two-dimensional quasicrystal with sub-nanometer resolution. But still doubt that this alone justifies publication in a high impact journal as Nat. Comm. (even if the data and the structure are beautiful). My requests to better highlight the new

insights into the physics of quasicrystals provided by this study and to go beyond a purely descriptive level were not really considered.

Thus, my feeling is that the authors unfortunately missed a chance to considerably improve this interesting manuscript in the first revision to make it suitable for publication in Nat. Comm.. If the open issues are addressed – including for example a tentative discussion of the locality – one could consider the manuscript for publication in Nat. Comm., otherwise I suggest publication in a more specialized journal.

Reviewers' Comments:

Reviewer #1:

Remarks to the Author:

With the amendments and additional experimental information the authors have added to the paper, the impressive quality and detail of the experimental observation becomes more apparent. However, at the same time the weakness of the discussion also becomes more apparent. From the analysis the authors present, it can be concluded that one should not view the CO/Cu(111) synthetic structures as a true realization of a TB vertex topology with edge hopping. It seems that the coherence of the resonances is of shorter range. I have mentioned this in my first report and authors suggest themselves this to be the reason why the B-sites show so little variance in the STS signatures. The authors take the example of the F and E site, having the same first order structure but showing different STS signatures, as an indication that aperiodic order still is important to influence the LDOS. However, this conclusion is not fully correct. The second or next nearest neighbor structure of the E and F site are very different. Whereas the F site has a rather uniform and dense 15 CO scatter ring at the second order, this feature is absent for the E site. I.e. the F site is kind of the center of a second order resonator ring. To some point it becomes a semantic discussion if a second (i.e. next nearest neighbor) order effect is already sufficient proof of the influence of aperiodicity. It certainly is not sufficient to study strict localization or critically of states in aperiodic systems.

Therefore it seems that the CO/Cu(111) surface state scattering is not really well suited to investigate coherent quantum mechanical effects of long range aperiodic order. Whereas I was just suggesting this in my first review, now I am convinced that this is the case. Nevertheless, the experiments are very interesting still, but rather than revealing some specific features of QM solutions of the Penrose tiling topology, they reveal aspects of the nature of the CO/Cu(111) scattering system. I think that the originality and quality of the experiments deserve publication in Nat. Comm., but the discussion must be honest and cannot claim unsupported conclusions.

I have issues with two similar sentences at the bottom of page 7 and page 8.

Page 7 "We must emphasize though that the electronic states are not being determined simply by the local potential created by the CO molecules. We can demonstrate this by showing how the E and F-sites have distinct electronic behavior."

The authors should discuss here the obvious strong difference of these sites regarding their second order structure. Such that a second order structure effect can be seen, but apparently it needs to be very strong to have an effect. This last statement is supported by the similarity of the signatures on the other first order vertex sites.

Page 8 "However, since the E and F-sites have the same local confinement as each other and exhibit different resonant states (Fig. 3), the local vertex structure and quasiperiodic order of the system are important in determining the electronic states."

This comes back to the fundamental contradiction, whether it is the extended aperiodic structure or the local structure dominates the resonances. The authors try to claim both. In my view, it is more appropriate to speak of a strong decay of the structural influence with the vertex order. Only where the second order structure differences are very particular (like for the E and F) it influences the LDOS. This is what the data shows and what would make a consistent discussion. Of course with the down side that the long range aperiodic order is of weaker influence. Also the FFT analysis is not really proving a higher order influence of the aperiodic structure. In a picture on non-interacting scatters the FFT would be the product of the reciprocal space pattern of the scatter positions multiplied with the structure factor of the single scatterer circular wave (i.e. ring at $2\cdot k(E)$), which is more or less seen by the authors. To see the multiple scattering effect in the FFT requires a very detailed investigation.

Further on Page 9

"The extension of this experimental approach to include quasiparticle scattering measurements could also solve other puzzles in the understanding of quasicrystals, such as whether the electronic states are localized or dispersive."

Having read this sentence several times I do not understand what the authors really mean. What type of quasiparticle are they referring to? How should this scattering reveal localization of dispersion?

In conclusion I think that the authors should very critically rethink (and accordingly discuss) how their experimental realization of an aperiodic QM system really does show long range aperiodic

structure effects or whether it is limited in this configuration only to local structure effect, which only in special cases extend to second order.

Reviewer #3:

Remarks to the Author:

In my opinion, the second revision again strengthened the manuscript and addressed my concerns, including the one about the locality of the electronic states. The wording was adapted, for example the comparison of the E and F sites is instructive. Furthermore, appropriate links to the completed Supplementary Information are now embedded in the manuscript. I'm still not enthusiastic about the novelty of the scientific conclusions and findings reported and thus still consider the manuscript as a borderline case for publication in Nature Communications. However, with the current status of the manuscript, I do not oppose acceptance and think that it can be published.

Reviewers' comments:

Reviewer #1 (Remarks to the Author):

With the amendments and additional experimental information the authors have added to the paper, the impressive quality and detail of the experimental observation becomes more apparent. However, at the same time the weakness of the discussion also becomes more apparent. From the analysis the authors present, it can be concluded that one should not view the CO/Cu(111) synthetic structures as a true realization of a TB vertex topology with edge hopping. It seems that the coherence of the resonances is of shorter range. I have mentioned this in my first report and authors suggest themselves this to be the reason why the B-sites show so little variance in the STS signatures. The authors take the example of the F and E site, having the same first order structure but showing different STS signatures, as an indication that aperiodic order still is important to influence the LDOS. However, this conclusion is not fully correct. The second or next nearest neighbor structure of the E and F site are very different. Whereas the F site has a rather uniform and dense 15 CO scatter ring at the second order, this feature is absent for the E site. I.e. the F site is kind of the center of a second order resonator ring. To some point it becomes a semantic discussion if a second (i.e. next nearest neighbor) order effect is already sufficient proof of the influence of aperiodicity. It certainly is not sufficient to study strict localization or critically of states in aperiodic systems.

Therefore it seems that the CO/Cu(111) surface state scattering is not really well suited to investigate coherent quantum mechanical effects of long range aperiodic order. Whereas I was just suggesting this in my first review, now I am convinced that this is the case. Nevertheless, the experiments are very interesting still, but rather than revealing some specific features of QM solutions of the Penrose tiling topology, they reveal aspects of the nature of the CO/Cu(111) scattering system. I think that the originality and quality of the experiments deserve publication in Nat. Comm., but the discussion must be honest and cannot claim unsupported conclusions.

We disagree with the referee that our experiment does not show long range order effects. In our experience building structures of much smaller sizes tends to give us very different results than larger structures, like our Penrose tiling quasicrystal. Although we don't have additional data to show you at the moment, we performed numerical simulations using scattering theory that to some extent, reproduces our data better than tight binding simulations do. In the figure below, we show you the simulated DOS spectra taken with increasing numbers of CO molecules around a B-site. In part a of the figure, we show the selection of CO molecules included in each simulated DOS spectrum, where the color-coded dots match up with the color-coded spectra in part b, and the B-site where we take the spectra is marked by a black dot. The spectra in part b have been offset for clarity, and starting at the bottom going up the radius of sites included in the simulation increases by 10 Å for each step. The scattering phase used in these simulations was chosen to best match data we have taken in the past with CO molecules.

Notice that until we reach more than 100 CO molecules, the spectrum continues to change, and the resonant peak near zero bias can't be distinguished with just nearest or second-nearest neighbors included in the simulation.

I have issues with two similar sentences at the bottom of page 7 and page 8.

Page 7 "We must emphasize though that the electronic states are not being determined simply by the local potential created by the CO molecules. We can demonstrate this by showing how the E and F-sites have distinct electronic behavior."

The authors should discuss here the obvious strong difference of these sites regarding their second order structure. Such that a second order structure effect can be seen, but apparently it needs to be very strong to have an effect. This last statement is supported by the similarity of the signatures on the other first order vertex sites.

Page 8 "However, since the E and F-sites have the same local confinement as each other and exhibit different resonant states (Fig. 3), the local vertex structure and quasiperiodic order of the system are important in determining the electronic states."

This comes back to the fundamental contradiction, whether it is the extended aperiodic structure or the local structure dominates the resonances. The authors try to claim both. In my view, it is more appropriate to speak of a strong decay of the structural influence

with the vertex order. Only where the second order structure differences are very particular (like for the E and F) it influences the LDOS. This is what the data shows and what would make a consistent discussion. Of course with the down side that the long range aperiodic order is of weaker influence.

Some aspects of our data could be understood from first and second order only, but that does not exclude the presence of long range order effects. We have changed the manuscript (highlighted on the bottom of page 7) to acknowledge that the difference in the spectrum of E and F could be coming from the second order structure. Since it is impossible to separate the effects of the local structure from the long range effects (which we hope the additional simulations of the B-sites with different numbers of surrounding sites is a sufficient demonstration of), we do not make a definite claim that one of the two is dominating. Instead, we indicate that the electronic behavior we observe in our system results from both the local structure and the aperiodic order. We changed the manuscript (highlighted on the bottom of page 8) to help clarify that both influences are important.

Also the FFT analysis is not really proving a higher order influence of the aperiodic structure. In a picture on non-interacting scatters the FFT would be the product of the reciprocal space pattern of the scatter positions multiplied with the structure factor of the single scatterer circular wave (i.e. ring at $2 \cdot k(E)$), which is more or less seen by the authors. To see the multiple scattering effect in the FFT requires a very detailed investigation.

We are not suggesting that the ring of radius $2 \cdot k$ is result of multiple scattering but it is simply an indicator of what the value of k is for the electrons at that energy. The important aspect is that in periodic systems, whenever $2 \cdot k$ matches the atomic peaks from the structure you get a standing-wave-like state that is resonant to tunneling. For example, in a square lattice this match happens at half filling, when you change from electron-like to hole-like dispersion. This is accompanied by a peak in the spectrum and a standing-wave-like state where the wavelength matches the lattice periodicity.

We asked ourselves if something similar would happen with quasicrystals. It is true that they don't have a well defined Brillouin zone but we produced the same match condition between the wavelength of the electrons and the brightest peak from the quasicrystal structure. Our result was the resonance at the B-sites which we talk about. We changed the manuscript (highlighted on page 6) to draw a stronger connection between the brightest peak and the standing wave resonance we observe. Notice once more that this resonance is not coming from first or second order confinement effects, as we show above.

Further on Page 9

"The extension of this experimental approach to include quasiparticle scattering measurements could also solve other puzzles in the understanding of quasicrystals, such as whether the electronic states are localized or dispersive."

Having read this sentence several times I do not understand what the authors really mean. What type of quasiparticle are they referring to? How should this scattering reveal localization of dispersion?

We apologize for using jargon from our field. We tried to clarify this sentence a bit more in the manuscript (highlighted on page 9). Fourier analysis of quasiparticle interference (QPI) has been used by STS groups as a way to map the band structure of many different materials. Extended states should show a well-defined momentum as a function of energy while localized states don't.

In conclusion I think that the authors should very critically rethink (and accordingly discuss) how their experimental realization of an aperiodic QM system really does show long range aperiodic structure effects or whether it is limited in this configuration only to local structure effect, which only in special cases extend to second order.

We hope we have convinced you in the examples above that the spectra in our system depends on more than second order, and we do indeed see long range aperiodic structure effects in our experiment.

Reviewers' Comments:

Reviewer #1:

Remarks to the Author:

With the changes made by the authors, I think that the presentation of the results and the discussion are apt for publication in Nature Communications.

I do agree that the (simulated) resonance pattern changes as a function of the patch radius up to a high order. This I would expect. The question however is rather, whether these changes depend on the aperiodic order on the outer regions, or just on the added density (i.e. depend on number not on specific arrangement) of scatterers? Not so much in the simulation, but in the experiment. I see however that the authors are working on this question, which I think is crucial to really understand the relation of the authors experiments and the theoretical models of the electronic structure of aperiodic systems.